# A Survey of Photoplethysmography and Imaging Photoplethysmography Quality Assessment Methods

**Théo Desquins** [1,2], **Frédéric Bousefsaf** [1,*], **Alain Pruski** [1,2] and **Choubeila Maaoui** [1]

1    LCOMS, Université de Lorraine, F-57000 Metz, France
2    I-Virtual, F-57000 Metz, France
\*    Correspondence: frederic.bousefsaf@univ-lorraine.fr

**Abstract:** Photoplethysmography is a method to visualize the variation in blood volume within tissues with light. The signal obtained has been used for the monitoring of patients, interpretation for diagnosis or for extracting other physiological variables (e.g., pulse rate and blood oxygen saturation). However, the photoplethysmography signal can be perturbed by external and physiological factors. Implementing methods to evaluate the quality of the signal allows one to avoid misinterpretation while maintaining the performance of its applications. This paper provides an overview on signal quality index algorithms applied to photoplethysmography. We try to provide a clear view on the role of a quality index and its design. Then, we discuss the challenges arising in the quality assessment of imaging photoplethysmography.

**Keywords:** contact photoplethysmography; imaging photoplethysmography; quality index

## 1. Introduction

A photoplethysmograph (also known as photoelectric plethysmograph, or PPG) is an instrument using light to measure blood volume changes in tissues [1,2]. A PPG signal can be extracted from several parts of the body (e.g., finger, ear, forehead, wrist) [2,3]. The conventional principle is called contact PPG (cPPG) and consists of a device in contact with the skin, emitting a controlled light beam. According to the Beer–Lambert law, this light will be transmitted, absorbed, or dispersed by the different elements constituting the tissues (e.g., blood components in capillaries or arterioles, keratin and melanin in epidermis [4,5]). Then, the output light is received by a photoreceptor [6]. The evolution with time of the light received by the receptor allows one to visualize the blood flow in a specific portion of tissue. The measurement of PPG can be based on the transmittance or reflectance mode. In transmittance mode, the source of light and the photoreceptor face each other and the receptor measures the light that has not been absorbed by the tissue. In reflectance mode, the light source and receptor are side by side and the receptor measures the light that is reflected from the tissue. PPG signals have applications in the monitoring of patients and extraction of vital signs [7].

The simplicity and low cost of the technology allow its integration in various devices, such as oximeters, smartwatches, or smartphones. It can be used in applications such as home monitoring, or in measurements performed during physical activities such as running.

Furthermore, the emergence of imaging photoplethysmography (iPPG) potentially allows the same applications using only the ambient light modulated and reflected by the blood from the skin and captured by a camera. The development of iPPG could allow physiological measurement during telemedicine, by using a webcam or a smartphone camera, to support the diagnosis of a clinician.

However, during a measurement, the signal obtained from a device can be corrupted by perturbations called artifacts. Consequently, the interpretation of the output signal is impossible or erroneous, which could lead to false diagnosis or to the rising of false alarms.

Thus, the discrimination between signals of high and low quality can help to detect the presence of perturbations. To avoid asking the opinion of an expert for each occurrence of an artifact and to maintain the high performance of the monitoring device, the field of quality assessment algorithms has been developed. Many algorithms have been suggested, evolving with the possibilities offered by devices, and depending on the application that the PPG signal is used for.

Several reviews have been presented on the topic of quality assessment of contact PPG signals. Nizami et al. [8] reviewed papers suggesting artifact detection algorithms for critical care units between 1989 and 2012. Orphanidou published a book [9] in 2018 addressing the quality assessment of contact PPG signals for rule-based and machine learning-based methods. Two smaller reviews have been recently published by Mejía-Mejía et al. [10] and Park et al. [11]. Mejía-Mejía et al. provided a definition of PPG signal quality and discussed methods used for its assessment. Park et al. published a review on PPG signal features, applications, and processing methods, including a section on signal quality indexes, in which they presented an overview of quality assessment algorithms oriented around feature-based methods and machine learning/deep learning-based methods. However, to the best of our knowledge, no survey has been done on the topic of quality assessment applied to iPPG. In this review, we try to provide an exhaustive view on the latest methods dedicated to quality assessment and discuss the methodology used for the creation of those algorithms for both cPPG and iPPG.

This paper presents an overview of the different methods used to assess the quality of a PPG signal and discuss the methodology in the design of those methods. Sections 2 and 3 of this paper provide the background on quality assessment, physiological signals, and photoplethysmography. Section 4 reviews the methods used for the quality evaluation of contact PPG over two differentiating points, the features extracted and the classification method. We finish by discussing the role and signal quality indexes (SQI) existing in iPPG.

## 2. Background

### 2.1. PPG Signal and Its Application

A PPG signal reflects the variation with time of the blood volume in the tissue of a patient [5]. The obtained signal is dependent on the mode (transmittance or reflectance) and on the wavelength of the light source used. Indeed, the light penetration depth in the tissue is different according to the wavelength used. Green light can only reach capillaries and upper arterioles under the epidermis, whereas red and infrared light are able to penetrate further into the dermis and reach deeper and larger arterioles [5]. This penetration depth has an influence on the quality of the signal under perturbation. Maede et al. [12] and Jihyoung et al. [13] showed that red- and infrared-based PPG are more subject to motion artifacts than green-based PPG. The PPG signal is composed of a semi-periodic component (AC component) reflecting the modification of blood flow due to cardiac activity and a baseline (DC component) varying with low-frequency physiological changes (e.g., respiration or variations of the sympathetic and parasympathetic system of the patient). The measurement of a PPG signal allows several applications [14]:

- Extraction of physiological parameters, which can be illustrated by the extraction or estimation of physiological variables such as pulse rate, pulse rate variability, blood oxygen saturation [15], blood pressure [16], jugular venous pulse, respiration rate, cardiac output, arterial stiffness, left ventricular ejection time [10].
- Monitoring of patients by the screening PPG signals to follow their cardiovascular state. It allows the rise of alarms in case of the detection of abnormal situations such as fibrillation in intensive care units.
- Diagnosis: for example, by the detection of cardiovascular or peripheral vascular diseases [17] or arrhythmia, such as atrial fibrillation [18] or ventricular tachycardia [19].

### 2.2. Measurement Factors Influencing a PPG Signal

This measurement of a PPG signal is a process influenced by five factors (Figure 1):

- the environment: the external conditions in which the measurement is taken (e.g., temperature, pressure, ambient light, occurrence of a perturbation from the environment hiding the portion of skin in case of iPPG);
- the material: the portion of skin subjected to the measurement;
- the operator: the person performing the measurement;
- the equipment: the device used for the measurement (e.g., a pulse oximeter/smartwatch for PPG or a camera for iPPG);
- the methods: the physics behind the measurement and the algorithm dedicated to the extraction of the desired measurand and the protocol of measurement.

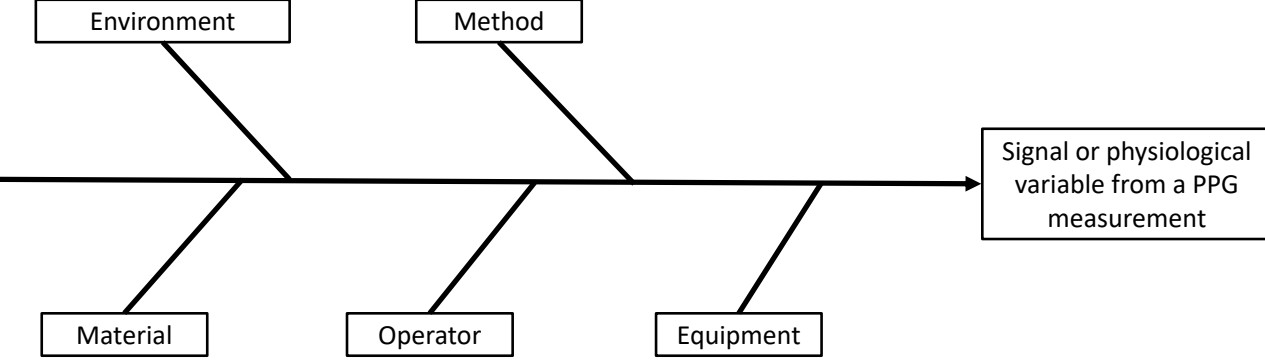

**Figure 1.** Ishikawa diagram presenting the factors influencing a photoplethysmographic measurement.

From each factor, perturbations can affect the quality of a PPG measurement (Table 1).

**Table 1.** Non-exhaustive perturbations affecting a PPG measurement.

| Measurement Factor | cPPG | iPPG |
|---|---|---|
| Means | Low resolution of the sensor<br>Inadequate sampling frequency of the sensor<br>Clipping<br>Sensor's noise | |
| | Power source interference [1] | Automatic exposition |
| | Contact pressure [20] | Rolling shutter [21] |
| | | Irregular frame rate [21] |
| Environment | Ambient light<br>Temperature [22] | |
| | | Low luminosity |
| | | Nonstable environment<br>(varying luminosity) |
| Material | Motion artifacts | |
| | | Presence of makeup [23] |
| | | Ballistocardiographic artifacts [24] |

### 2.3. Physiological Factors Influencing a PPG Signal

In addition to measurement perturbations, several studies have highlighted the influence of physiological factors (e.g., skin tone, skin thickness, arterial thickness, venous return) and states of the patient (such as illness, age, weight, blood pressure) on the shape of a PPG signal (see Table 2).

**Table 2.** Physiological factors affecting a PPG signal.

| Physiological Factor | cPPG | iPPG |
|---|---|---|
| Measurement site | Impact on the shape of the PPG pulse [25] | |
| Autonomic nervous system (sympathetic and parasympathetic) | Impact on the baseline of the signal [26] | |
| | Impact on pulse amplitude [7,27,28] | |
| Arterial stiffness | Impact on the waveform [29,30] | |
| | Impact on pulse wave velocity [29,30] | |
| Presence of disease (Arrythmia, premature ventricular contraction, etc.) | Pulse to pulse interval [1,18] PPG waveform [7,18] | Pulse with multiple peaks (diabetes), incomplete pulses (arrythmia) [31] |
| Respiration | Baseline wandering Modification of the waveform shape [25] Modification of frequency and amplitude of the AC component [29] | Modification of the pulse rate [32] |
| Skin thickness | Lower PPG signal intensity and modification of the waveform [29] | |
| Skin tone | Higher absorption of lower-wavelength light (green) [29] | Signal of lower amplitude and more subject to noise [33] |
| Venous return | Affects both low-frequency components and AC components [29] | |

## 3. Introduction to Signal Quality Index for Photoplethysmographic Signals

### 3.1. Definition of Quality

We distinguish between two types of quality for photoplethysmography, the metrological quality and the physiological quality.

#### 3.1.1. Metrological Quality

Metrological quality focuses on the measurement factors affecting both the photoplethysmographic signal retrieved and the physiological parameters extracted from it. This aspect of the signal quality is subject to perturbations (detailed in Table 1) arising from the environment, the operator (e.g., motion artifacts, poor installation of the sensor for contact PPG), the sensor (e.g., inadequate sampling frequency or resolution), or the mean. Different perturbations can occur and impact the signal with varying intensities, such as different amplitudes of a motion artifact or noise power of a sensor. Thus, we represent the notion of the quality of the signal with a continuous scale (Figure 2). This scale represents how much the quality of the retrieved signal is affected and depends on the method implemented to extract and process the signal, such as the performances of an algorithm of face tracking, signal extraction, and filtering for an iPPG measured from the face. For example, a smartwatch application dedicated to the measurement of photoplethysmographic signals during exercise may be more subject to motion artifact than a pulse oximeter device used at rest. However, the use of motion artifact reduction algorithms [34] allows one to recover a signal from some perturbations. This processed signal can ultimately be labelled as good-quality. Metrological quality focuses on the following question: given a measurement

with a protocol and method, in a given condition of measurement, how well does the obtained signal reflect the variation in blood volume in the tissue?

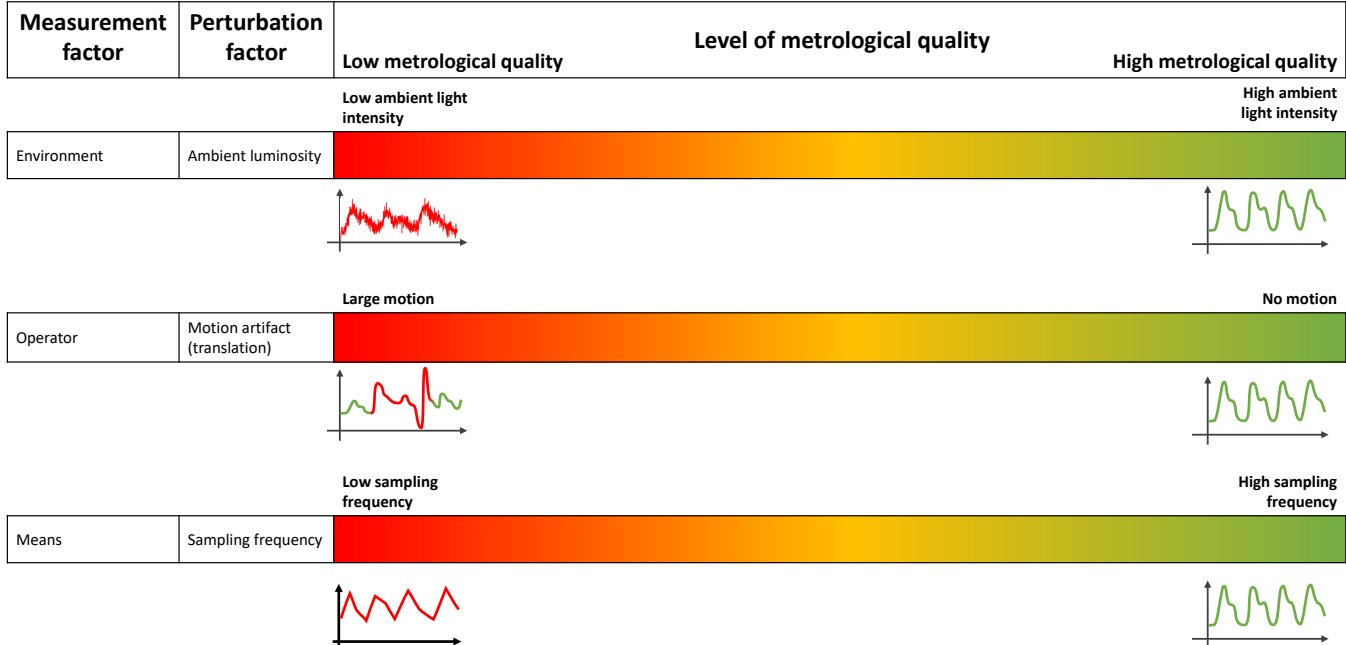

**Figure 2.** Non-exhaustive example of perturbation factors influencing the metrological quality. For each perturbation, the color bar represents the metrological quality of the signal. Signals tending to the red color have low metrological quality and will have high metrological quality when tending to the green color. Depending on the intensity of the perturbation factor, it has an influence on how well the signal retrieved from the sensor reflects the variation in blood volume in tissues.

### 3.1.2. Physiological Quality

The notion of physiological quality is specific to the nature of the PPG signal. The measurement of PPG is done on living tissues. Consequently, the obtained signal is affected by physiological factors. For example, let us consider a PPG signal measured in optimal conditions with a perfect sensor and a proper processing algorithm, giving a signal whose variations are only due to blood volume changes in tissues, supposing also that we want to use this signal to measure the pulse rate. Then, the signal retrieved may still not be suitable for this application if the patient suffers from arrhythmia. Here, we want to highlight that, given some applications built upon the PPG signal (e.g., estimation of pulse rate, blood oxygenation, or pulse rate variability), the quality of the signal can be independent of metrological factors. Physiological factors can affect the PPG signal with different intensities. For example, an arteria can be more or less stiff or the respiration intensity of the patient can vary. Consequently, we also decided to represent physiological quality with a continuous scale (Figure 3). This notion of physiological quality is dependent on the application of the PPG signal and the algorithms implemented to achieve it (including the signal extraction process).

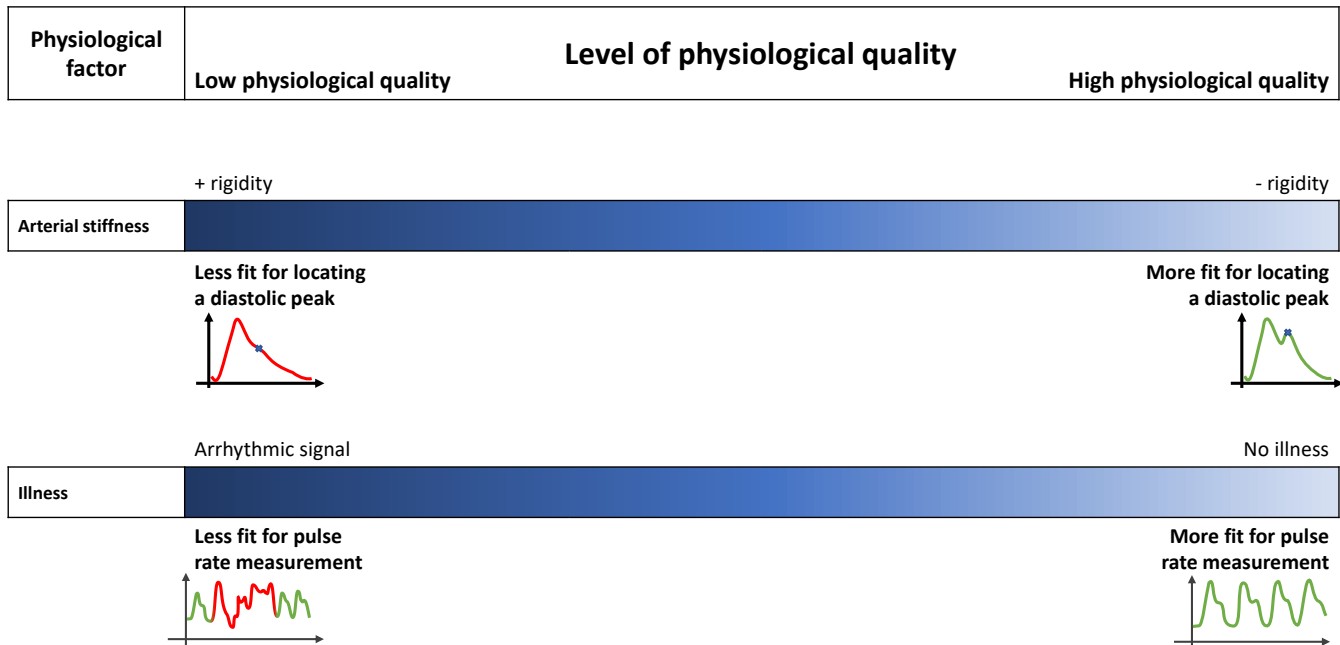

**Figure 3.** Non-exhaustive example of physiological quality illustrated with the waveform of a fingertip contact PPG. The physiological factors presented here are not measurement perturbations. Here, the case of optimal measurement conditions with a perfect sensor is represented. The signal reflects the variation in blood volume. However, the shape of the obtained signal cannot be used for its application.

### 3.1.3. Annotation

From the above sections, quality has been defined as how well the signal translates the variation in the blood flow of a patient with the metrological quality and how well the signal can be used for the derived application with physiological quality using continuous scales. In practice, quality is commonly addressed as discrete. Most of the reviewed papers presented in this survey associate the quality of the PPG signal binarily ("good" or "bad" quality) or with intermediate discrete levels. Passing from two continuous notions of quality with respect to metrological and physiological factors to a discrete representation makes the annotation of a set of signals difficult. However, it is a necessary step for the evaluation of the performance of an SQI algorithm.

**Automatic annotation**

Automatic annotation processes refer to annotation methods without the intervention of a human operator.

A common method is to extract a physiological variable (e.g., PR, stroke volume) from the PPG signal and compare it to the value of the same physiological variable extracted from another device and/or another signal (e.g., electrocardiogram (ECG)) serving as a reference. If the difference between the values does not exceed a threshold, then the PPG signal is considered of good quality; otherwise, it is considered of low quality. In this case, the notion of quality is principally driven by the application.

Another method consists in annotating the signals with the help of an already existing SQI. This method is used for the case of papers wanting to deploy an SQI algorithm replacing one or several older SQI algorithms.

**Manual annotation**

Manual annotation involves human intervention in the process. As with any manual annotation, this process takes time and is difficult. It is also subject to errors from annotators (noise in labels).

The annotation of PPG signals differs from the annotation of objects in images [18] as follows.

- A PPG signal is a physiological signal, and evaluating its metrological quality needs the opinion of experts in the field. Consequently, the process of annotation can be expensive, and the annotation of large datasets is difficult.
- Because of its physiological nature, a PPG signal measured from a patient must be coherent with the physiological possibilities of the human body and with other physiological signals (e.g., ECG or continuous blood pressure). This coherence offers the possibility for the annotators to be guided
    - with a set of rules that the PPG signal must comply with;
    - with another physiological signal not affected by the perturbation.

Other than subjective or guided annotation, a third way to manually annotate the quality has been proposed. This method consists in including artificial perturbations during the recording of the PPG. For example, the luminosity can be changed during the recording of an iPPG signal, or a patient may be asked to move his/her hand at a given time (Figure 4) during the recording of the cPPG. Then, the PPG signal corresponding to the occurrence of the perturbation is labelled as low-quality. The impact of the perturbation on the signal may still depend on the intensity of the artifact induced. Table 3 proposes a summary of the different methods of annotation.

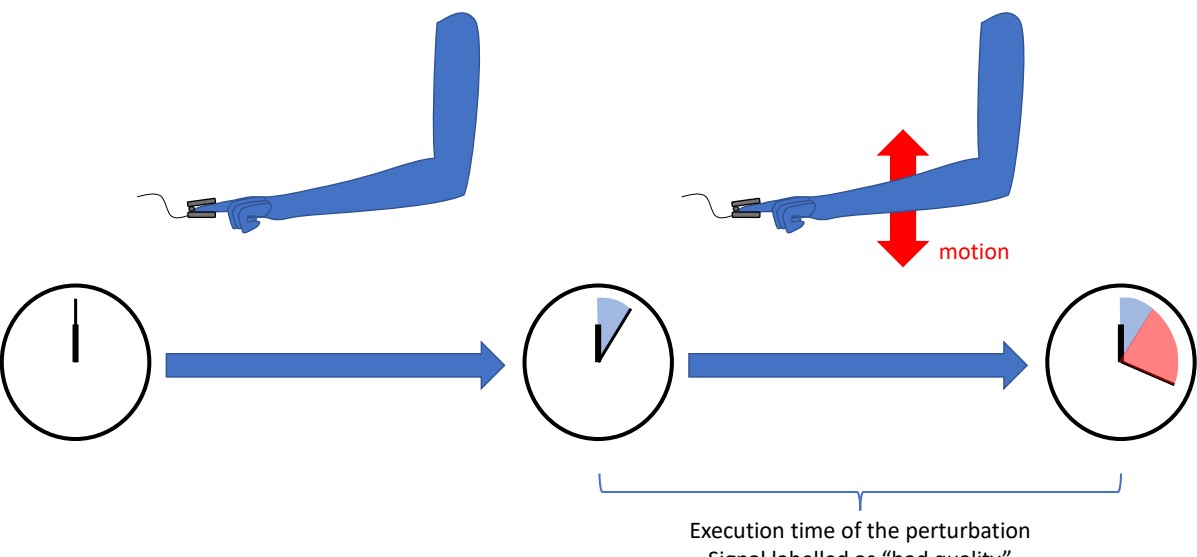

**Figure 4.** Example of annotation done during a controlled experiment. Here, a patient is wearing a PPG sensor and is asked to not move during a given period. Then, at a specific time, the patient is asked to artificially create a perturbation (here, a motion artifact with up–down movement). The signal recorded during this period of perturbation is automatically annotated as "bad-quality".

**Table 3.** Classification of papers addressing SQI according to the annotation type.

| References | Details | | Annotation Type |
|---|---|---|---|
| [35] | Evolution of PR between 2 pulses and sudden evolution of the value of the blood oxygenation. | | Automatic annotation |
| [35–38] | Comparison between PR estimated from the PPG signal and the HR estimated from ECG. | | |
| [39,40] | Comparison between stroke volume measured from cPPG and a reference taken from impedance cardiography. | | |
| [40] | Comparison of left ventricular ejection time measured by a reference with the one measured by cPPG. | | |
| [41] | Labelling by computing already existing quality indicators. | | |
| [36,42–50] | Not Guided | | Manual annotation |
| [51–54] | Guided | By rules | |
| [55,56] | | By reference signal | |
| [35,57–59] | Perturbations artificially created during recording | | |

PR = Pulse Rate, HR = Heart Rate, ECG = Electrocardiogram, cPPG = Contact Photoplethysmography.

## 3.2. Applications

We define as a quality index an algorithm taking a PPG signal in entry and assigning it a score or label indicating its quality. A signal quality index can be used in several applications:

- Reducing false alarms during patient monitoring;
- Presenting clean signals for experts' interpretation;
- Cleaning datasets for machine learning applications;
- Suppressing irrelevant signals to maintain performance of physiological variable predictions;
- Integrating the SQI into the signal processing algorithm to evaluate or improve its performance.

The general purpose of an SQI consists in increasing the reliability of the application in which it is implemented. We can distinguish two ways to use an SQI: as an indicator for keeping or removing a signal and as an indicator to improve the performance of the signal processing chain.

### 3.2.1. Keeping or Removing PPG Signals

The main reasons to remove a set of PPG signals are as follows:

- Avoiding misinterpretation and false alarms from corrupted signals;
- Maintaining the performance of an application by keeping only relevant signals;
- Preparing an artifact-free dataset for the training and testing of a machine/deep learning estimator to achieve good performance [60].

In the case of cleaning, a signal quality assessment algorithm can be used as a tool to indicate which signal should be kept and which should be suppressed (see Figure 5).

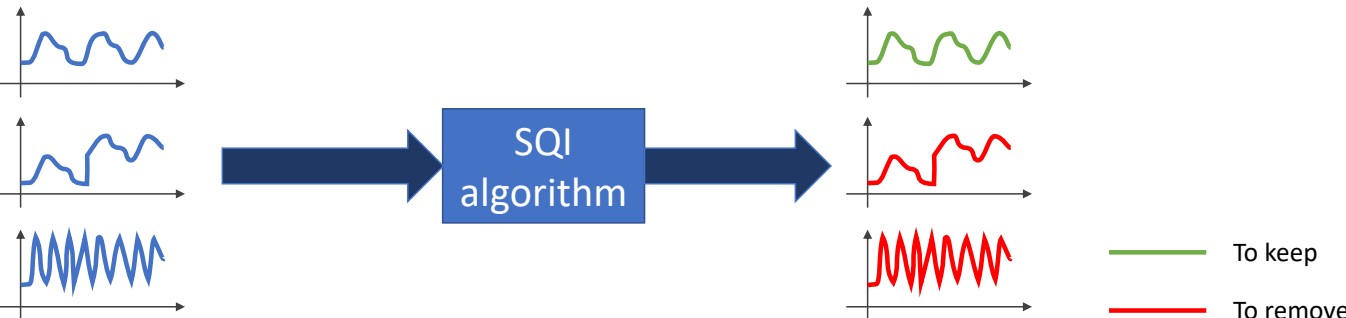

**Figure 5.** Example of application of an SQI assigning a quality score or label to filter irrelevant data. In this figure, an SQI assigns a "bad-quality" label (in red) for a noisy PPG signal or a signal affected by a motion artefact to be suppressed and a "good-quality" label (in green) for the signals to be kept in the dataset.

An SQI algorithm can be evaluated for its contribution to the performance of the estimation. This kind of evaluation is application-oriented. However, it may not represent accurately how the SQI algorithm behaves as it does not consider the proportion of good signals unnecessarily removed or the proportion of bad signals still present after cleaning.

Some applications need a more accurate evaluation of the signal. For example, in the case of false alarm reduction, it in necessary to separate a corrupted signal from an episode of arrhythmia. This second method relies on an annotated database. The performance of the SQI algorithm is evaluated by comparing the estimated signal quality with the annotated reference.

The SQI evaluation metrics are usually taken from a confusion matrix (accuracy, sensitivity, precision, positive predictive value (PPV), Area Under Curve–Receiver Operating Characteristic (AUC-ROC)). Accuracy gives the global performance of the SQI algorithm, while sensitivity and precision focus, respectively, on the proportion of signals well classified within the real bad-quality signals and on the proportion of correctly estimated samples in all estimated bad-quality signals. AUC-ROC is the area under the curve in the graph relating the sensitivity and false positive rate obtained by moving the threshold between the good and bad signals. An algorithm with an AUC close to 1 is able to separate the good from the bad signals.

3.2.2. Supporting the Signal Processing Chain

In this scenario, the algorithm can be integrated into the processing algorithm. For example, in [32,61], a remote iPPG signal is extracted from an image divided into several subregions. Each subregion provides an iPPG signal and these signals are evaluated by an SQI algorithm (signal to noise ratio (SNR) in this case). The score obtained for each signal is then used to weight the contribution of each subregion to form the final iPPG signal. An SQI algorithm can also be used to evaluate different signal extraction and processing methods. For example, Wang et al. [62] compared different methods of extraction of imaging PPG (PCA, ICA, CHROM [63], Spatial Subspace Rotation (SSR) [64], and Plane Orthogonal to Skin (POS) [62] by evaluating the quality of the signal extracted from a common database using an SNR quality index.

## 4. SQI for cPPG

We grouped quality assessment algorithms for cPPG into three categories: rule-based, machine learning-based, and deep learning-based method.

### 4.1. Rule-Based

In this section, we focus on the SQI algorithm as a set of criteria computed from the PPG signal. These criteria rely on the expected behavior or on specific properties (resulting from statistical analysis). These criteria reflect morphological features (see Section 4.1.1

and Figure 6). The SQI consists here in verifying if their values or variations are within a plausible physiological range. These physiological ranges are defined from a statistical study from the literature or datasets. They can also consist in the detection of events unlikely to have a physiological origin, such as clipping [44,45,65,66], i.e., truncation due to the saturation of the signal. The stability of the PPG waveform is also an important property of the signal, used as a criterion to evaluate its quality.

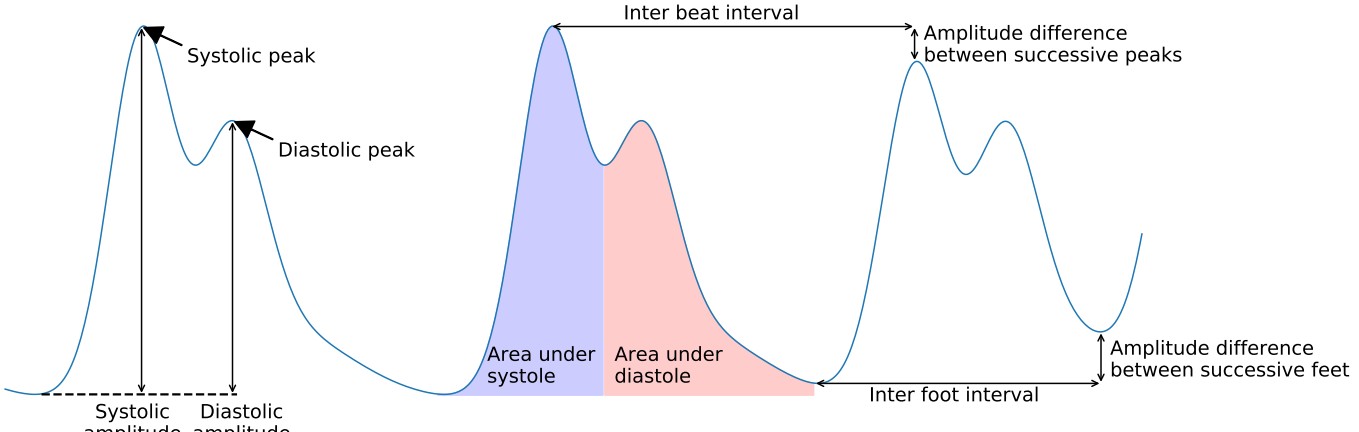

**Figure 6.** Figure illustrating cPPG pulses and possible features extracted from them.

### 4.1.1. Criteria Based on Signal Features

To address the different criteria reviewed in papers suggesting a rule-based SQI algorithm, we decided to group them within three scales. Signal scale criteria regroup conditions based on morphological and spectral statistics computed on an ensemble of successive pulses (see Table 4). Pulse scale criteria regroup conditions based on morphological and spectral features computed for each pulse. Fischer et al. [44,45] suggested to evaluate the quality of a PPG pulsation based on the value taken by its morphological features (pulse width, rising time, pulse amplitude, area under the pulse, difference between autocorrelation of red and infrared signals). Each threshold separating good- from bad-quality signals has been designed from the literature and on statistical analysis. Among features computed on pulses, the skewness of the pulse is the most commonly used. This feature indicates the asymmetry of a PPG waveform and should be positive as the systolic peak is dominant in a normal pulse. Mohamed Elgendi demonstrated in [51] that skewness is the best indicator to separate excellent-quality from medium- and low-quality signals. The inter-beat scale regroups conditions based on the variation in features between successive beats. Within inter-beat features, the inter-beat interval (the time separating two successive pulse peaks) is the most commonly used feature and gives an idea of the stability of the signal from one beat to another.

In Table 4, we only enumerate features extracted from a PPG signal. However, if the device used to measure the PPG signal offers measurement from other sensors, such as the acceleration from a smartwatch accelerometer [67], the use of features from other signals is often observed to assess its quality. For example, Sappia et al. [48] used the red and infrared PPG signals offered by their sensor to deduce the variation in oxygenated and deoxygenated blood and build criteria based on it (ratio between oxygenated and deoxygenated pulse). These criteria are not applicable for smartwatches measuring PPG from green light.

**Table 4.** Enumeration of the different PPG signal features extracted for the assessment of its quality.

| Scale | Domain | | Features | References |
|---|---|---|---|---|
| Pulse | Temporal | Amplitude | Pulse amplitude | [44,45,68,69] |
| | | | Area under pulse | [44,45] |
| | | | Mean, std of pulse waveform | [35,37] |
| | | Temporal | Pulse width | [44,45] |
| | | | Pulse rate | [36,70] |
| | | Derivative | Ratio of maximum positive slope over minimum negative slope | [59] |
| | Spectral | | Entropy | [51] |
| | | | Principal frequencies and residual noise | [69] |
| | Statistical | | Skewness | [37,50,51,57,69] |
| | | | Kurtosis | [37,50,51,57,69] |
| Inter-beat | Temporal | | IBI | [35,37,46,59,71] |
| | | | Successive IBI | [59] |
| | | | Inter-foot interval | [46] |
| | | | Amplitude difference between successive peaks (mean, std) | [35,37,44–46] |
| | | | Amplitude difference between successive feet | [46,69] |
| | | | Difference between pulse widths | [44,45] |
| | | | Difference between rising times | [44,45] |
| Signal | Temporal | | Amplitude of the signal | [48,72] |
| | | | Difference between autocorrelation of PPG from red and infrared | [48] |
| | | | Ratio of oxygenated and deoxygenated blood measured | [48] |
| | | | Ratio between systolic and diastolic time | [35] |
| | | | Kurtosis | [58] |
| | | | Shannon entropy on signal amplitude distribution | [47,58] |
| | | | Shannon entropy on signal amplitude distribution | [47,58] |
| | | | Permutation and sample entropy | [73] |
| | | | Predictor (autoregressive model) coefficients fitted on 5 s signals | [66] |
| | Spectral | Variable-Frequency Complex Demodulation (VFCDM) | Residual noise | [36] |
| | | | Projected frequency modulation difference | [36] |
| | | | Difference between PR and IBI | [36] |
| | | | Energy and amplitude of dominant frequencies in PR range | [74,75] |
| | | | Energy of non-dominant frequencies in PR range | [74] |
| | | | Variation of dominant frequencies with time | [75] |
| | | Power spectrum | Variation of dominant frequencies with time | [37] |
| | | | Spectral entropy | [73] |
| | | | Hjort parameters | [37] |
| | Statistics | | SNR | [47,51] |
| | | | Variance of the signal | [37] |
| | | | 2 first peaks of correlogram of the signal | [47] |
| | | | First peak amplitude and time and number of zero crossing of the auto-correlation of the signal | [72] |
| | | | Detrended Fluctuation Analysis, Fractal Dimension, and Higuchi Fractal Dimension | [73] |

IBI = Inter-Beat Interval, PR = Pulse Rate, SNR = Signal to Noise Ratio, std = Standard Deviation.

### 4.1.2. Stability of the PPG Waveform

Among the reviewed papers, some suggest using the stability property of the waveform of the PPG signal to discriminate between good- and bad-quality signals. Instability of the waveform of a PPG signal's pulsations is likely due to measurement perturbations such as motion artifacts or sensor clipping. To check the stability of a PPG pulse, a common process is to extract the pulses from the PPG signal. Then, each pulsation is compared to a reference, i.e., a model of pulsation built from the signal and called a template. If the pulse is different from the reference, the pulse is considered as low-quality. For example, Orphanidou et al. [70] computed a template as the mean pulsation of a portion of signal neighboring the pulsation to be evaluated, whereas Karlen et al. [43] used only the previous pulsation as the template. John et al. [76] built a template iteratively within the first few seconds of the signal. The authors first spot an interesting portion of a pulse (with points of interest) and take it to initialize the template. Then, they iteratively use the Toeplitz correlation matrix to find the next portions of pulse aligned with the template and add it to the current template. On this principle, several papers have suggested original methods to build a pulsation model and compare it with the PPG pulse to be evaluated. Building a template of a pulse as the average of detected pulses in the signal [43,46,65,70,77,78] or using neighbors [43,79] is a common method. The difference is based on how the pulses are selected for the building of the template (Li et al. [65] used all pulses in a 30 s windows, while Karlen et al. [43] preselected the pulses by performing normalized cross-correlation) and how the signals are aligned ([46,77,78] applied dynamic time warping (DTW) for the alignment of the pulses that [80] used to build the template; John et al. [76] first spotted a portion of signal presenting interest points and then iterated over the signal to find aligned portions of signals to add to the template). Comparing the template with new pulses is mainly done with the correlation [43,65,70,79] and the DTW distance [53,65,77,78]. Sabeti et al. [46] made a comparison using the Kullback–Leibler divergence, while John et al. [76] suggested a more original way to compare the signals by first computing a point-to-point difference between the template and the signal and then computing the variance of the obtained signal. If the variance is high, then the quality is low. Lim et al. [81] suggested a comparison with several pulses, in a model called the master template. These templates are obtained from the two first components of a PCA analysis performed on pulses of the MIMIC II dataset [82]. A pulse is then compared with the template maximizing normalized correlation. If the pulse and its neighbors are not labelled as artifacts, then it is used to update the template.

The rule-based methods present several limitations. As the different conditions are commonly designed manually, another statistical study may be needed when the SQI algorithm is applied to a PPG signal obtained by a different process of measurement (e.g., different device and conditions of measurement). Moreover, the performance of the algorithm of the SQI depends on the signal processing and feature extraction algorithms and should be evaluated along with the SQI algorithm [45]. The reviewed methods are summarized in Table 5.

**Table 5.** Summary of the rule-based type of quality assessment algorithms.

| Reference | Data | Performance |
|-----------|------|-------------|
| [68] | 13 subjects<br>8 records of 1 min per subject | Cohen's $\kappa$ = 0.64<br>Sen = 0.89<br>Spe = 0.77<br>Acc = 0.83 |
| [58] | 24 subjects<br>>134 min of recording | Acc = 0.888<br>Sen = 0.869<br>Spe = 0.983 |
| [43] | Capnobase [83]<br>Complex System Laboratory [84] | 90% of the signals with artifacts had a score<br>below 95/100<br>14% of good-quality signals are labelled as<br>artifacts for a score below 85/100 |
| [65] | MIMIC II [82] | Evaluation by the impact on false alarm<br>reduction with suppression |
| [70] | Physionet/CinC 2011<br>+ author-collected database | Sen = 0.91<br>Spe = 0.95 |
| [44,45] | 63 subjects<br>31.5 h of annotated signals | Performance of extended algorithm:<br>Acc = 0.984<br>Sen = 0.995<br>Spe = 0.916<br>Pre = 0.986 |
| [77] | Capnobase [83]<br>Complex System Laboratory [84] | Performance on Capnobase<br>for a threshold of 0.8:<br>Sen = 0.9664<br>PPV = 0.9926 |
| [81] | 19 subjects (>5 min per record)<br>+ PhysioNet MIMIC II | Mean performance (over authors' dataset):<br>Acc = 0.935<br>Sen = 0.869<br>Spe = 0.902 |
| [53] | 3 subjects<br>6 min records | Best performance:<br>Acc = 0.9258<br>Sen = 0.9297<br>Spe = 0.9218<br>PPV = 0.9225 |
| [48] | 14 subjects<br>158 records of 10 s | Classification performance:<br>Acc = 0.9268<br>Sen = 0.9286<br>Spe = 0.9245<br>Pre = 0.6420<br>F1-score = 0.9353 |
| [79] | Capnobase [83]<br>Complex System Laboratory | Sen = 0.9466<br>PPV = 0.9678 |
| [66] | 15,000 records of 5 s from 3 different devices<br>+ MIMIC II [82] + Complex System<br>Laboratory [84] + Wrist [85] + Cup [86] | Acc = 0.9321<br>Sen = 0.9822<br>Spe = 0.9071 |
| [72] | 19,700 segments of 4 s | Acc = 0.9989<br>Sen = 0.9994<br>Spe = 0.9939 |

Acc = Accuracy, Pre = Precision, Sen = Sensitivity, AUC = Area under Curve, PPV = Positive Predictive Value.

### 4.2. Machine Learning

As with rule-based algorithms, their performance is dependent on the features selected and the algorithms implemented to extract them. Selecting the features for quality level separation adaptable to PPG signals from different datasets needs further design effort.

Couceiro et al. [69] extracted 26 features from the period-domain short-time Fourier transform and performed feature selection via normalized mutual information. This technique shows how knowing the value of a feature reduces the uncertainty in the prediction of the quality of the signal while keeping only features independent from each other. In this study, they extracted four temporal and four periodic features. Sabeti et al. [46] focused their efforts on the computation of features that would not be affected by the change of PPG device. The authors first computed reference values of features and templates with the first few seconds of each record. Then, they used them to normalize the features computed from the signal and feed machine learning models to predict the quality.

Contrarily to rule-based techniques, the use of machine learning has the advantage of avoiding statistical studies to design the thresholds separating the different levels of quality. It also allows complex non-linear combination of the signal features. Support vector machine (SVM) is the most commonly used model among publications [36,46,69,87,88]. It has the ability to non-linearly project the extracted features to a higher-dimensional space to increase the probability of finding a hyperplane separating good from bad signals. The SVM algorithm has become a reference found in many of the comparisons described in SQI papers. Sabeti et al. [46] described a benchmark over several machine learning methods (SVM, classification and regression tree, ensemble decision tree, and threshold optimization) and found that SVM offers less overfitting and maintains high performance over several datasets. They also compared their method using SVM with two other algorithms [43,77] on the Capnobase dataset [83]. Pereira et al. [52] also found the best performance with their SVM classifier against seven representative methods [42,43,68,70,77,89,90]. Elgendi used the SVM algorithm to find the most discriminant feature to classify the quality of a PPG pulse [51]. Dao et al. [36] compared their method with a smaller SQI including an SVM using time-domain features. Pereira et al. [55] compared the performance of different classic deep learning architectures but included the SVM in their comparison.

Designing supervised learning methods for the quality assessment of a PPG signal makes the step of signal annotation necessary (whose issues are highlighted in Section 3.1.3) and adds the difficulty of hyperparameter tuning. For example, Li et al. [42] performed an ablation to find the best parameters of their network. The authors tested the use of either four inputs (three correlations with a template and clipping detection) or six inputs (four previous entries with number of detected pulses and a combination of the four previous entries) and the number of neurons in the hidden layer of a multi-layer perceptron (MLP). A summary of the different SQI based on machine learning methods is presented in Table 6.

### 4.3. Deep Learning

Deep learning methods have been employed in the field of PPG signal quality assessment. Feature extraction is integrated in the architecture of a deep neural network (DNN) [38]. It allows one to regroup several quality assessment algorithms into one deep learning architecture [41,54,91] and gives higher freedom for the input passed to those networks [40,54,55,92].

Figure 7 illustrates typical methods applied to the quality assessment. Roh et al. [54] converted PPG pulses into 2D images using a recurrent plot and then classified each pulse into good or bad quality using two 2D convolutional neural networks (CNNs) with max dropout and a dense layer with softmax activation (Figure 7a). Azar et al. [92] decomposed each 6 s PPG signal into 78 approximation coefficients using DWT. A 1D autoencoder (with CNN and bidirectional Long Short-Term Memory (LSTM) layers) is trained to reconstruct clean PPG signals. Because the network is only trained on clean signals, the reconstruction of the signal fails for PPG with bad quality and the mean square error between the input and the rebuilt signal is high. If this error is over a threshold, then the signal is considered as bad-quality (Figure 7b). Figure 7c illustrates the method implemented by Guo et al. [56]. The authors trained an autoencoder (UNet with residual blocks) to perform quality assessment as a segmentation task. They assign a score between 0 and 1 to each point of a 30 s PPG signal, which allows a precise estimation of the quality. Then, they evaluate the performance of their trained network with the DICE score (ratio between the intersection of portion of signal estimated and annotated as artifact over the portion of signal estimated or annotated as artifact).

**Table 6.** Summary of machine learning methods dedicated to quality assessment.

| Reference | Data | Machine Learning Method | Performance |
|---|---|---|---|
| [42] | 104 subjects, 1055 pulsations | MLP | Acc = 0.952<br>Sen = 0.990<br>Spe = 0.806<br>PPV = 0.952 |
| [35] | 33 subjects | SVM + temporal neighbor voting | Mean performance on 3 artefacts<br>(finger motion, head motion, walking):<br>Acc = 0.938<br>Sen = 0.943<br>Spe = 0.924 |
| [69] | 15 subjects<br>22 records of 1 min per subject | C-SVC | Mean performance:<br>Acc = 0.885<br>Sen = 0.843<br>Spec = 0.915 |
| [36] | 5 different datasets<br>(Chon Lab and UMass Medical center) | SVM | Precision in the detection of the<br>occurrence time of a MNA<br>Difference in Transit Time = 0.91 ± 0.59 s |
| [52] | 13 subjects | SVM | Acc = 0.9033<br>Sen = 0.9505<br>Spe = 0.9163 |
| [37] | 17 subjects<br>24 h record per subject | SVM | Acc = 0.984<br>Sen = 0.8550<br>Spe = 0.9184 |
| [59] | 40 subjects<br>records of 1.5 to 2 min | Fuzzy neural network | Mean performance:<br>Acc = 0.8992<br>Sen = 0.8421<br>Spe = 0.9363 |
| [46] | 46 subjects<br>+ Capnobase dataset | Test of 3 machine learning models:<br>Classification and regression tree, SVM,<br>ensemble tree | Best score from SVM, mean performance:<br>Sen = 0.9576<br>Spe = 0.9190<br>PPV = 1 |
| [47] | 26 subjects | Test of k-nearest neighbor,<br>multi-class SVM, Naïve Bayes,<br>decision tree, random forest | Best score for random forest:<br>Acc = 0.745 |
| [39] | 10 subjects<br>3 min per record | Fuzzy neural network | Performance on detection of<br>bad-quality pulses :<br>Acc = 0.86<br>Pre = 0.97<br>Sen = 0.84 |
| [73] | 30 subjects<br>10 min per subject | Self-organizing map | Acc = 0.9201<br>Sen = 0.9580 |
| [50] | 5 subjects<br>12 min of recording per hour for<br>each subject during 6 days | Unsupervised elliptical envelope<br>algorithm | Results of the leave-one-subject-out test<br>in the classification of bad quality:<br>Pre = 0.85<br>Sen = 0.98 |

Acc = Accuracy, MLP = Multi-Layer Perceptron, MNA = Motion Noise Artifact, Pre = Precision, Sen = Sensitivity, PPV = Positive Predictive Value, SVM = Support Vector Machine, C-SVC = C-Support Vector Classifier.

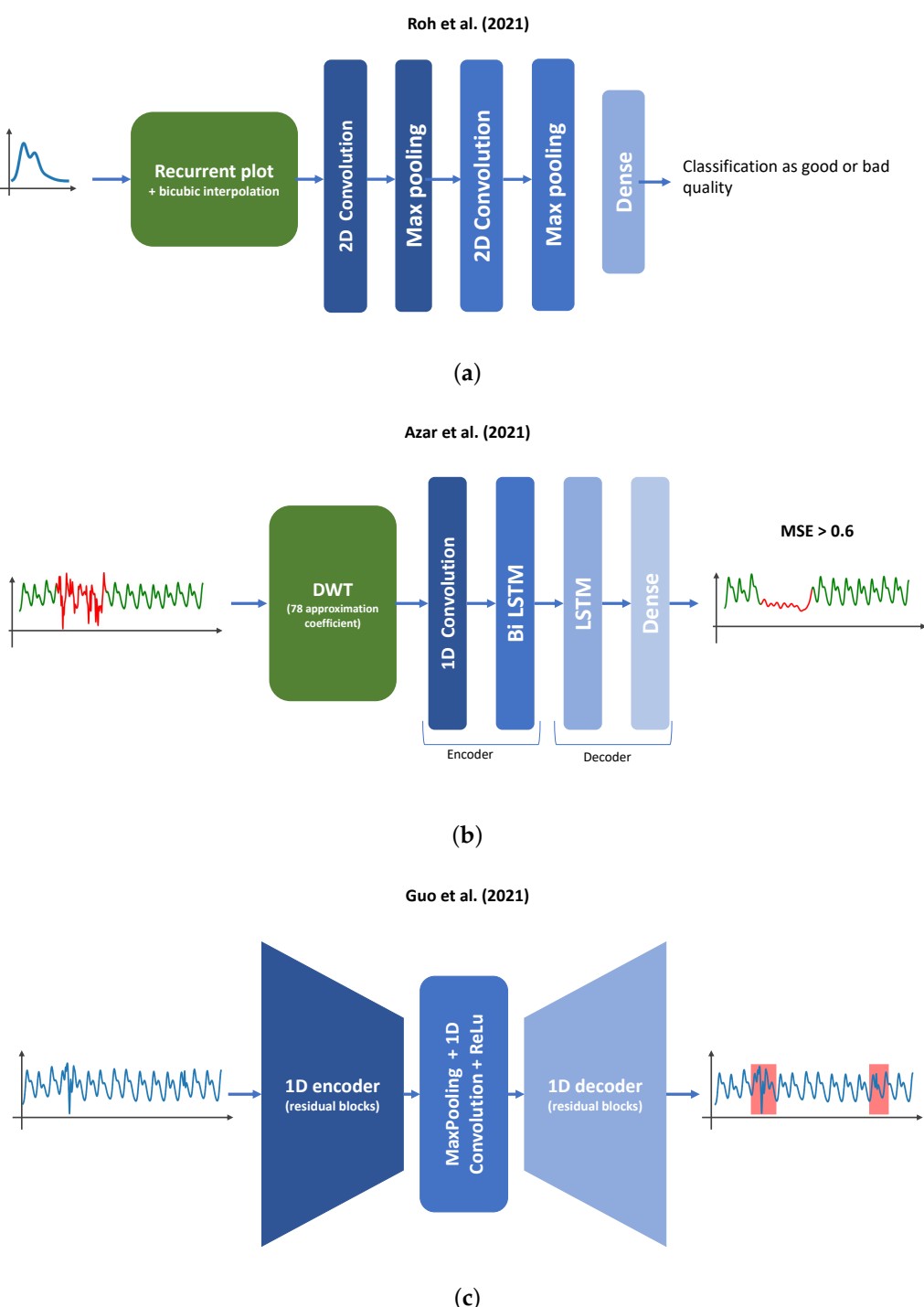

**Figure 7.** Presentation of several deep learning architectures dedicated to quality estimation of a cPPG signal. (**a**) Architecture used by Roh et al. [54] using 2D convolution to treat each pulse of the signal as an image. (**b**) Architecture implemented by Azar et al. [92]. Because the architecture is only trained on clean PPG signals, it will fail to reconstruct a PPG signal when the input is corrupted, thus increasing the MSE between the reconstructed signal and the input signal. With a manually chosen threshold, it allows one to make the distinction between good and bad signal quality. (**c**) Architecture implemented by Guo et al. [56] showing the estimation of the quality of the signal as a segmentation problem using deep learning.

Signal processing is still necessary for the generalization of deep neural networks over several datasets. Moreover, this kind of SQI algorithm needs a large dataset with annotated data for training, which is difficult and expensive in the case of manual annotation (see Section 3.1.3). Automatically annotating datasets with already existing SQI allows one to solve the problem of the number of samples but without any relevant increase in performance. Another main limitation of the use of deep learning is the lack of transparency in the features learned in the PPG signal fed to the DNN. To tackle this issue, Zhang et al. [93] suggested to evaluate the explicability of two DNN artifact detectors (a 1D Resnet-34 and a 2D Resnet-18) with the attention score offered by the DeepSHAP, Integrated Gradient and Guided Saliency. A summary of the different deep learning-based methods can be found in Table 7.

**Table 7.** Summary of deep learning-based quality assessment algorithms.

| Reference | Data | Entry | Deep Learning Architecture | Performance |
|---|---|---|---|---|
| [67] | 19 subjects 1443 records of 30 s length | 30 s PPG signal + index indicating motion detected by the smartwatch accelerometer injected into the dense part of the network | 3 CNN + 2 dense layers | Acc = 0.9002 AUC = 0.9521 |
| [38] | 5 days data collection | 60 s signal window | 1 CNN + dense network | Best performance: AUC = 0.88 Pre = 0.7674 Sen = 0.8354 |
| [55] | 2 private datasets | 2 entries are tested: 1D entry, 30 s of normalized PPG signal 2D entry, normalized RGB plot image of the signal | 1D entry: Attention LSTM ending with dense network Fully connected network 2D entry: VGG19, Resnet18, Resnet50, Xception | Best performance (ResNet18): Acc = 0.9851 Spe = 0.9791 Sen = 0.9877 |
| [40] | 14 subjects | 2D images of the plot of zero-padded PPG pulse and its derivative | ResNet50; VGG19 | Best performance (ResNet50): Acc = 0.94 Pre = 0.96 Sen = 0.92 |
| [49] | 183 subjects | Raw PPG signal | 1 CNN + 1 shared CNN + 2 separated CNN with dense layer | - |
| [91] | 38 subjects BIDMC and ICU dataset from MIMIC | 5 s of normalized PPG signal | 13 layers 1D CNN | Acc = 0.945 Sen = 0.967 Spe = 0.904 |
| [54] | 76 subjects | PPG pulse signal converted to recurrence plot | 2D CNN + 1 dense | Acc = 0.975 Sen = 0.964 Spe = 0.987 |
| [41] | PPG : DaLiA dataset [94] and Cuffless Blood Pressure Estimation dataset [95] iPPG: 6 subjects | 5 points window of 64Hz PPG signal | 3-unit LSTM + dense layer | Average Acc = 0.7973 |
| [92] | 2 subjects + data augmentation | Discrete wavelet transform (DWT) approximation coefficients from second level of 6 s PPG signal | Autoencoder used Encoder : 1D CNN + bidirectional LSTM layer Decoder : LSTM layer + dense layer | Pre = 0.90 Sen = 0.95 |
| [56] | 44 subjects | 30 s bandpass-filtered signal | 1D U-Net (5 residual encoder, 5 residual decoder) | Mean performance DICE score = 0.8734 ± 0.0018 |

Acc = Accuracy, CNN = Convolutional Neural Network, LSTM = Long Short-Term Memory, Pre = Precision, Sen = Sensitivity, AUC = Area under Curve, PPV = Positive Predictive Value.

## 5. Design of SQI for iPPG

### 5.1. Existing Studies

Imaging PPG is the photoplethysmography measurement using a camera. Several reasons motivate research in this technology:

- A camera is a widely used technology allowing remote and continuous monitoring of physiological variables (e.g., PR, PRV);

- This technology imposes less constraints on the patient as the signal can be taken remotely;
- Several potential applications exist for telemedicine (e.g., PR, blood pressure, blood oxygenation).

However, this kind of measurement is done in a less controlled environment [96] and is also subject to measurement perturbations (see Table 1). Even if iPPG is still an emerging technology, signal quality indexes have been used to guide the extraction methods implemented to retrieve the signal from the camera [63,97,98].

Botina-Monsalve et al. [99] compared the quality of their iPPG signal extracted using periodic variance maximization [97] after filtering it by a bandpass filter, a wavelet filter, or a deep learning architecture using SNR and a template matching method [70] as metrics. Gao et al. [41] designed a 1D signal quality index for PPG and iPPG using a deep LSTM architecture. They highlight the difficulty of designing deep learning methods for iPPG quality assessment as the proportion of bad-quality signals is important in iPPG, which unbalances their training dataset. Moreover, the training of deep learning architectures is difficult as the number of iPPG datasets available for research is limited. The authors augmented their initial dataset with artificial data to train their model.

Bobbia et al. [98] considered the idea that an iPPG signal extracted from different regions of the face may have different qualities. The authors introduced unsupervised superpixel segmentation, to segment the image of the face of a patient into different regions of interest. The final iPPG signal is obtained by summing the PPG signals extracted from each superpixel weighted by their respective SNR. Benezeth et al. [61] extended the work of Bobbia et al. [98] by lowering the SNR complexity. The authors first trained a Hidden Markov Model (HMM) to build a probabilistic distribution modeling the evolution of an iPPG signal. The authors also created a HMM to model the noise of the signal. The SNR is then computed as the ratio between the likelihood that the evolution of the signal is explained by the HMM modeling the iPPG signal and the likelihood that it is explained by the HMM modeling the noise. Fallet et al. [100] used the average value of the absolute difference between the tracked forehead pixel amplitude at frame $n$ and $n-1$. The authors observe sudden increases in its value when a perturbation, such as motion or an illumination change, occurs. They also set different strategies of artifact detection according to the iPPG extraction method. The authors added an additional step for iPPG signals extracted by POS [62] and SSR [64] methods (as these methods are more resistant and react differently to perturbations) by detecting if a change in the amplitude of the signal (characteristic of a corrupted signal) happens in a suspicious portion of the signal. Tables 8 and 9 provide respectively a summary of the features used to design SQIs for imaging PPG and of the existing methods.

**Table 8.** Features used for the evaluation of iPPG.

| Scale | Domain | Features | References |
|-------|--------|----------|------------|
| Pulse | Temporal | Amplitude — Difference between systole and diastole (pulse amplitude) | [101] |
| | | Amplitude — Amplitude before and after perturbation for iPPG extracted by POS or SSR | [100] |
| | | Template matching [70] | [99] |
| Signal | Temporal | Std of the signal | [64,101] |
| | Spectral | SNR (from frequency spectrum) | [63,97–99,101,102] |
| | | Relative difference between highest and second highest amplitude of the signal spectrum | [101] |
| | | Maximum scalar product between PPG periodogram [103] and predefined filters | [104] |
| | Probabilistic | SNR (obtained using HMM models) | [61] |

HMM = Hidden Markov Model.

**Table 9.** Papers dedicated to signal quality indices for iPPG.

| Reference | Data | Algorithm Type | Performance |
|---|---|---|---|
| [100] | 31 video records | Rule-based algorithm | Evaluated on contribution over HR estimation |
| [61] | UBFC-RPPG dataset [105] | Hidden Markov Model | Evaluated on contribution over HR estimation |
| [104] | 200,000 smartphone PPG records | Rule-based algorithm | - |
| [36] | 5 different datasets (Chon Lab and UMass Medical Center) | SVM | Evaluated on contribution over HR estimation |
| [106] | 226 subjects | Fitting each pulse to a sinusoidal model using non-linear least square optimization. If the fitting fails (i.e., no convergence or error high) or the model parameters are outside a statistical range, the pulse is bad-quality. | Evaluated on impact on error between HRV estimated from PPG and ECG |
| [101] | Bingamton–Pittsburgh–RPI Multimodal Spontaneous Emotion database [107] | Rule-based algorithm | Evaluated on contribution over HR estimation |
| [41] | 6 subjects + augmentation with PPG DaLiA dataset | 3-unit LSTM + dense layer | Evaluated on contribution over HR estimation |

HR = Heart Rate; LSTM = Long Short-Term Memory, SVM = Support Vector Machine.

iPPG and cPPG share common properties such as semi-periodicity and the extractable features of their respective pulses. Some SQIs can be either used for cPPG or for iPPG. For example, approaches based on the signal to noise ratio [51,98] can either be applied to cPPG or iPPG signals. The difference lies in the threshold value separating good from bad signals. Some deep learning methods can either be applied to iPPG or cPPG. For example, Gao et al. [41] developed a single LSTM architecture for the assessment of cPPG and iPPG signals. Both types of signals can be analyzed with SQI developed for 1D signals. The case of iPPG is particular as the signal is computed from images. SQI applied to iPPG can be computed on two-dimensional matrices, thus allowing one to analyze subregions of the image [98].

*5.2. Potential Developments in This Domain*

The creation of SQI for the measurement of iPPG comes with new challenges. As mentioned above, iPPG presents significant potential for telemedicine-based applications. However, to democratize this technology, its measurement must be reliable despite the high diversity of uncontrolled measurement contexts and perturbations (see Table 1). To overcome these difficulties, the implementation of an SQI is a solution to maintain the performance of these applications.

In this regard, the use of a camera can be regarded as an opportunity in the design of an innovative SQI. This sensor allows one to extract information on the context of measurement and offers the possibility to design a more complete SQI. The exploitation of contextual information could allow one to distinguish perturbation factors linked to the metrological and physiological aspects of the quality of the signal. For example, Wang et al. [108] created a signal quality index for iPPG by first designing indices for the quality of the context of measurement and then for the quality of the signal extracted. Within the quality of the measurement context, the authors considered the environment of the measurement by computing indices related to the intensity of the light source, the light spectrum and direction, the type of light source (punctual or parallel source of light), and the quantity of skin exposed to the measurement. The authors also considered a physiological factor, the skin tone, in the study of their SQI. The use of camera also allows one to pass to a spatio-temporal notion of quality. Indeed, the quality of an iPPG signal extracted may be uneven over the different skin regions [109] and may also change with time. Only a few

papers [61,98] take into account the spatial dimension of the quality. Thus, the field of SQI applied to iPPG has room for improvement and could be a sandbox for new signal quality algorithms.

## 6. Conclusions

In this review, we have presented an overview of the existing methods for contact photoplethysmography and imaging photoplethysmography. We have subdivided the different quality assessment algorithms according to their use of designed rules from statistical studies or from the use of machine learning/deep learning algorithms. We have also reviewed the new challenges set by the emerging iPPG and its potential applications. As we believe that iPPG will further develop, we have tried to encourage the development of new methods of quality estimation in this field by suggesting new lines of research to be explored in this field.

**Author Contributions:** Review of the literature, T.D. and F.B.; writing—original draft preparation, T.D., F.B. and A.P.; supervision, F.B., A.P. and C.M. All authors have read and agreed to the published version of the manuscript.

**Funding:** This research received no external funding.

**Institutional Review Board Statement:** Not applicable.

**Data Availability Statement:** Not applicable.

**Conflicts of Interest:** The authors declare no conflicts of interest.

## Abbreviations

The following abbreviations are used in this manuscript:

| | |
|---|---|
| CNN | convolutional neural network |
| DTW | dynamic time warping |
| DWT | discrete wavelet transform |
| HMM | hidden Markov model |
| HR | heart rate |
| HRV | heart rate variability |
| ICA | independent component analysis |
| LSTM | long short-term memory |
| MLP | multi-layer perceptron |
| PCA | principal component analysis |
| POS | plane orthogonal to skin |
| PPG | photoplethysmography |
| cPPG | contact photoplethysmography |
| iPPG | imaging photoplethysmography |
| PR | pulse rate |
| SNR | signal to noise ratio |
| SQI | signal quality index |
| SSR | spatial subspace rotation |
| SVM | support vector machine |

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
