# Peer review of "A Survey of Photoplethysmography and Imaging Photoplethysmography Quality Assessment Methods"

_applsci, doi:10.3390/app12199582_

Round 1

Reviewer 1 Report

The work is written very nicely.

I recommend an unconditional acceptance to the paper.

Reviewer 2 Report

This paper reviews the signal quality assessment algorithms for PPG and iPPG. This paper divided the PPG SQI into rule-based, machine learning-based, and deep learning-based. SQI for iPPG is also reviewed. New challenges and its potential applications of iPPG are discussed. This paper needs the following improvements.

1. Other existing papers reviewing the signal quality assessment for PPG and iPPG must be included. The difference between this review paper and existing papers must be discussed.

2. SQI should be defined when it first appears.

3. In background, the quality is defined by metrological quality and physiological quality. However, in the later review, the methods are not discussed in terms of metrological quality and physiological quality. The association should be enhanced.

4. In Section 5, the differences of cPPG SQI methods and iPPG SQI methods should be analyzed. Are they the same? If no, what makes the differences?

5. In conclusion, I do not quite understand the term “probabilistic tools”. Please explain.

6. Typos and grammatical errors exist. A thorough spell check should be performed.

7. Most importantly, I suggest the authors to implement some representative methods and compare the performance. This enables direct comparison between algorithms on a consistent experimental platform, which will be contributing to the research community.

Reviewer 3 Report

The manuscript by Desquins and colleagues provides a review of the methods employed to assess the quality of photoplethysmography (PPG) signals. Overall, the manuscript is interesting and written well enough. The authors present information in a concise way in Tables and Figures that are easy to consult and understand. However, some sections, particularly the Introduction, would benefit from several corrections:

1) I suggest that authors start to distinguish the two types of PPG techniques - reflection and transmission - which should come with a brief explanation of the basis of functioning. Please include a mention to the different wavelengths used and how they might influence signal quality. Finally, please emphasize that PPG is used only to measure skin and/or skeletal muscle perfusion, depending on the wavelength;

2) I also suggest that authors include a figure of the normal PPG waveform and the main parameters that can be extracted from it, for the benefit of readers who are not fully aware of the PPG potential;

3) Line (L) 17: The authors use the term "blood" (tissue) with "keratin" and "melanin" (substances). I suggest that "blood" be replaced by "blood components" or by names of specific components of blood;

4) L24: it would be interesting to state which arrhythmias can PPG detect;

5) L44 and 46: please uniformize the term that you want to use - either "paper" or "article";

6) L49: please define SQI;

7) L56 and Table 2: please be very careful when mentioning the influence of the parasympathetic nervous system. Since most (if not all) PPG measurements are made on the skin, which does not receive parasympathetic innervation, the only way that nervous branch can affect PPG waveforms is by changing pulse rate;

8) L56-64: these are almost a repetition of the potentials by PPG that were described in the Introduction. I suggest to suppress the paragraph from the Introduction and keep the one from the Background section;

9) L70-71: the "material" source of perturbation seems to be more an anatomical/physiological factor rather than a measurement factor. Please revise;

10) L73: I suggest a different term for "means", perhaps "devices";

11) L81: what is meant  by "mass"?

12) Table 2: this table does not associate a specific change in the specific factors to their effect on PPG waveform. I also suggest to include "anatomical site" as a "physiological" factor affecting the PPG signal since a waveform may be interpreted as of bad quality or signifying disease if the anatomical site is not mentioned/known when in fact it can be simply from an atypical location;

13) L99-105: please provide a reference for the study(ies) where authors are able to collect "clean" PPG signals during motion/exercise;

14) Figure 2: in the middle left graph the PPG line is over the yy axis. Please correct. The same applies for Figure 3, graph b);

15) Figure 3: a PPG waveform compatible to what the authors define as reflecting higher arterial rigidity can be identified in atypical locations in healthy subjects with non-rigid arteries;

16) L138: what "threshold" do the authors refer to?;

17) L158-163: please do not forget that even if artificial perturbations are introduced in the PPG signal for reference, they are still dependent on the intensity. I suggest the authors include that comment in the manuscript;

18) L197: please define AUC/ROC;

19) Table 1 and L225: please define "clipping";

I also recommend the reviewers an English proofread of their manuscript in order to improve readability and remove some typos. 

Round 2

Reviewer 2 Report

The authors responded to my concerns. Although the suggested comparison was not conducted, some references that have done comparisons were cited instead. This manuscript can be accepted. 

Reviewer 3 Report

The authors have satisfactorily answered all my questions, having provided sound justifications. Their manuscript has now enough quality for publication.

I only have a minor comment regarding Figure 6 - I suggest rewriting the caption; also in the figure itself, perhaps "Amplitude difference between successive feet" instead of "Amplitude difference between successive foots" would be more correct.